# Multimodal Graph Networks for Compositional Generalization in Visual Question Answering

**Raeid Saqur**[1,2,3*]
[1]University of Toronto Computer Science
[2]Princeton University, Computer Science
[3]Vector Institute for Artificial Intelligence
`raeidsaqur@cs.[toronto|princeton].edu`

**Karthik Narasimhan**
Department of Computer Science
Princeton University
`karthikn@cs.princeton.edu`

## Abstract

Compositional generalization is a key challenge in grounding natural language to visual perception. While deep learning models have achieved great success in multimodal tasks like visual question answering, recent studies have shown that they fail to generalize to new inputs that are simply an unseen combination of those seen in the training distribution [6]. In this paper, we propose to tackle this challenge by employing neural factor graphs to induce a tighter coupling between concepts in different modalities (e.g. images and text). Graph representations are inherently compositional in nature and allow us to capture entities, attributes and relations in a scalable manner. Our model first creates a multimodal graph, processes it with a graph neural network to induce a factor correspondence matrix, and then outputs a symbolic program to predict answers to questions. Empirically, our model achieves close to perfect scores on a caption truth prediction problem and state-of-the-art results on the recently introduced CLOSURE dataset, improving on the mean overall accuracy across seven compositional templates by 4.77% over previous approaches.[2]

## 1 Introduction

In this paper, we explore the problem of systematic generalization to novel questions in the paradigm of visual question answering (VQA) [4]. Several neural architectures have shown great promise in learning multimodal representations to solve the task [42, 39, 54]. Recently, neuro-symbolic methods [55, 38] – a combination of connectionist and symbolic approaches – have pushed these limits, achieving close to perfect scores on benchmarks like CLEVR [28, 29]. However, these models lack the ability to generalize to new combinations of linguistic constructs, even if the distribution of visual inputs remains the same [6]. A key reason behind this is the lack of fine-grained representations that allow for joint compositional reasoning over both visual and linguistic spaces.

We propose a graph-based approach for multi-modal representation learning – the *Multimodal Graph Network* (MGN) – with an explicit focus on enabling better generalization. Our core idea is the following: representing both text and image as graphs naturally allows for tighter coupling of concepts between the two modalities and provides a compositional space for reasoning. Specifically, we first parse both the image and text into individual graphs with object entities and attributes as nodes and relations as edges. Then, we induce a *correspondence factor matrix* between pairs of nodes from both modalities using message passing algorithms similar to the ones used in graph neural networks [16]. The output of our model is a matrix of values that captures the correspondence value of each pair of

---

[*]Work done at Princeton as a Fulbright Scholar.

[2]Code is available at `https://github.com/raeidsaqur/mgn`

individual nodes between modalities. A high-level overview of our approach is shown in Figure 1, where one can observe fine-grained connections between nodes from graphs of both modalities.

Our work is different from previous approaches to multimodal learning for VQA since we explicitly parse and represent both modalities. This provides two key advantages – 1) it allows for learning explicit correspondences between visual concepts and linguistic symbols at a fine-grained level, and 2) our representations scale smoothly to longer pieces of text (e.g. questions) with novel compositions of linguistic constructs. Our entire model is end-to-end differentiable and can be trained using supervision provided in the end task (e.g. classification or VQA).

We evaluate MGN on two tasks – a binary classification task of predicting if a caption matches an image based on attribute compositions in the CLEVR dataset [28], and CLOSURE [6] – a recently released challenge for testing systematic generalization in language. We show the efficacy of our approach with strong compositional reasoning skills, achieving a $\approx$98% average accuracy rate on statement truth prediction on previously unseen image feature combinations at test time. Further, on CLOSURE, our model outperforms state-of-the-art approaches by almost 4.77% overall accuracy, especially achieving impressive results on compositional questions entailing logical (`and,or`) objet relations.

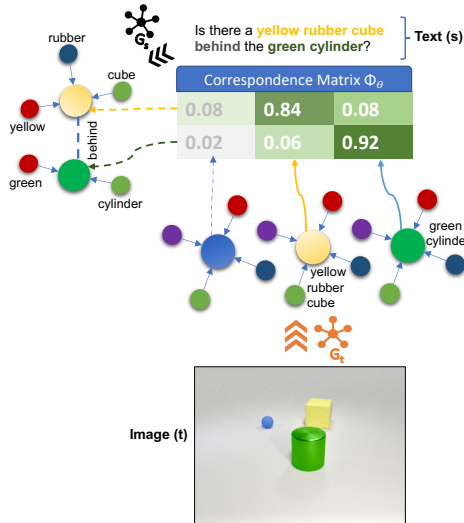

Figure 1: Overview of our graph-based approach to fine-grained multimodal representation learning. We parse both modalities and the model induces a *correspondence matrix* between every pair of nodes to represent similarity.

## 2   Related Work

**Neuro-symbolic approaches to VQA**   There have been several recent attempts at merging symbolic reasoning with neural networks to boost compositional generalization in models for visual question answering. Neural module networks [3] parse questions into substructures and dynamically construct a neural network conditioned on these components. NS-VQA [55] and NS-CL [38] learn explicit semantic representations over images and sentences, using direct and indirect supervision, respectively. These representations are used to generate and execute symbolic programs to answer questions. While we also construct explicit representations, our approach uses probabilistic factor graphs and graph neural networks to encourage a tighter coupling between concepts in the two modalities.

**Graph-based approaches to VQA**   Several recent papers has explored the use of graphs for multimodal vision and language tasks. Santoro et al. [42] and Chang et al. [9] employ Relation Networks, which can be thought of as a basic form of graphical reasoning over pairs of nodes, and demonstrate very good performance on VQA tasks like CLEVR. Hu et al. [25] also propose a graph-based representation over objects in an image and use message passing algorithms to update object representations conditioned on text (e.g. questions). Teney et al. [47] use graphs to represent both images and text but merely combine the representations using a weighted sum. The main difference in our work is that we represent both visual and textual inputs as graphs and perform a soft graph matching to map nodes between the two modalities. Concurrent to this work, Gao et al. [15] also proposed a fused multimodal representation, using iterative message passing between text and visual embeddings for the grounded SCAN task [41].

**Generalization in VQA**   The *VQA-CP* and *VQA-2.0* [2, 19] datasets, and corresponding models (e.g. GVQA [2], which builds on *stacked attention networks* (SAN) [54]) test *generalization* across distributional shifts in train-test class labels. Transformer [48] based models like SAN, LXMERT, ViLBERT [46, 37] use a two-stream architecture to embed each modality separately, then fuse them together using attention-based interactions. These approaches are orthogonal to MGN in problem setting and architecture and could potentially be combined. The CLEVR dataset already incorporates

non-uniform train-test labels, and SAN performs reasonably well (73.2%) on CLEVR (baseline in NS-VQA). However, SAN and, by extension, GVQA architectures do not evaluate for, and generalize poorly on, unseen object attributes (CLEVR-CoGenT) and linguistic structural pattern (CLOSURE) combinations. While the *Neural State Machine* (NSM) model [26] also builds a scene graph from the image similar to our approach, it requires converting the question text into a sequence of instructions for graph traversal. In contrast, we convert the text input into a graph as well and perform multi-modal fine-grained matching.

Recent work has explored models with the explicit aim of enabling compositional reasoning and generalization. [33] propose a challenging dataset for few-shot generalization based on written alphabets. [35] proposed a task of translating natural language instructions into explicit action sequences, with the goal of composing terms like 'jump' and 'twice' to intrepret compound instructions. Within the realm of visual QA, [28] proposed a variant of the CLEVR task that involved a held out set of unseen combinations of concepts. However, their test is limited to the visual modality since the combinations swapped were on image object features – it doesn't test language compositionality. CLOSURE [6] is a more recently proposed dataset based on CLEVR that aims to test systematic generalization of VQA models to questions unseen at training time. They created several challenging question templates that combine atomic constructs used in CLEVR, and construct a new model, Vector-NMN to handle the challenges.

## 3 Multimodal Graph Networks (MGN)

We first provide the motivation behind our approach. Consider the image in Figure 1 and the associated question: '*Is there a yellow rubber cube behind the large green cylinder*'. Answering this question requires first locating the green cylinder and then scanning the space behind it for a yellow rubber cube. Specifically, 1) although there may be other objects present (like another ball for e.g.), information about them can be abstracted away and 2) fine-grained connections between the visual and linguistic inputs representing 'yellow' and 'cube' need to be established. Neural architectures that build multimodal representations using convolutional neural networks (CNNs) process the entire image into a single global representation (e.g. vector), but fail to capture such fine-grained correlations [29]. Neuro-symbolic approaches like neural module networks [3] or NS-VQA [55] mitigate this to some extent by using multiple filters to process the image, each conditioned on different relations extracted from the text. However, these approaches fail to scale well to longer compositions of linguistic constructs [6].

To satisfy both desiderata, we induce graphical representations over both modalities, which allows for tighter coupling between fine-grained concepts (e.g 'yellow', 'cube'). Specifically, we parse both image and text into two different graphs and then use a neural message passing algorithm [16] to jointly reason over them and induce *factor correspondences* between each pair of nodes in the disjoint graphs. Finally, we use a graph-based aggregation mechanism to generate a multi-modal vector representation.

### 3.1 Model

Assume each multi-modal input instance to be a tuple $(s, t)$ where $s$ is a *source* text input (a question or caption for e.g.), and $t$ is the corresponding *target* image. Our Multimodal Graph Network (MGN) architecture is made up of two main components: **1.** a **Graph Parser**, and **2.** a **Graph Matcher**.

#### 3.1.1 Graph Parser

The graph parser takes the input instance $(s, t)$, and outputs corresponding object-centric graphs $G_s = (V_s, A_s, X_s, E_s)$ and $G_t = (V_t, A_t, X_t, E_t)$. Here, $X \in \mathbb{R}^{|V| \times d}$ is the feature matrix of all the nodes $V$ in graph $G$, $A$ is the adjacency matrix, and $E \in \mathbb{R}^{|E| \times d}$ is the feature matrix of all the edges $E$ in graph $G$.

For the input text $s$, the parser uses an *entity recognizer* [40] to capture the objects and attributes as graph nodes $V$, followed by a *relation matcher* [40] to capture node relations as edges in $G_s$.

For an image $t$, the parser uses a pretrained Mask-RCNN [22, 17], and ResNet-50 FPN [36] image semantic segmentation pipeline [23] to obtain objects, attributes and positional coordinates $(x,y,z)$. These form individual nodes in graph $G_t$.

Does the `metal cylinder CLEVR_OBJ` in `front SPATIAL_RE` of the `yellow rubber object CLEVR_OBJ` have the `same color MATCHING_RE` as the `small rubber ball CLEVR_OBJ` ?

Following the construction of nodes and edges in $G_s$ and $G_t$, the embedding matrices $X$ and $E$ are obtained by using word embeddings from a pre-trained language model (LM) as feature vectors of dimension $d$ over the graph nodes (object, attributes) and edges (relations). Similarly, for the image scene graph, we use the (object, attributes) labels obtained from the 'parsed scenes' (from Mask-RCNN pipeline) as input to the shared LM to get feature embeddings. Figure 2 illustrates the input and output flow of the parser conceptually.

### 3.1.2 Graph Matcher

The *graph matcher* takes as inputs the graphs $G_s = (V_s, A_s, X_s, E_s)$ and $G_t = (V_t, A_t, X_t, E_t)$ produced by the *graph parser* and generates a multimodal vector representation $\vec{h}_{s,t} \in \mathbb{R}^{2d}$ – this captures a latent joint representation of the source nodes (in text) and matching target nodes (in image).

We use $h_i^{(0)} = x_i \in X$ as our initial representation of the nodes across both graphs, and use a shared graph neural network (GNN), $\Psi_\theta$, to compute localized, permutation equivariant [20, 7] node representations. At the end of this neural message passing, each node encapsulates information from its surrounding neighbors.

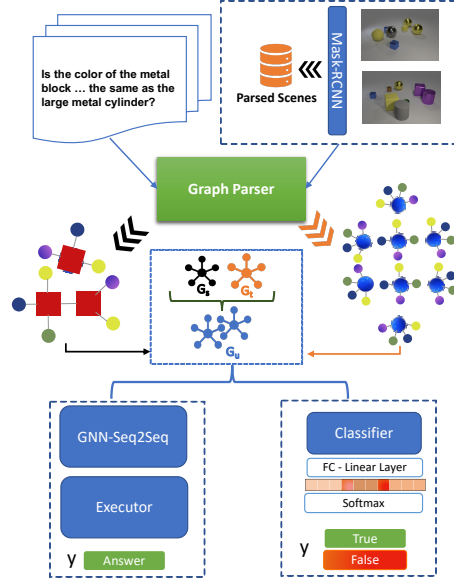

Figure 2: **MGN Architecture**: Both image and text are parsed into graphs $G_t$ and $G_s$ by the *graph parser*. The *graph matcher* combines them and runs joint neural message passing to obtain a correspondence matrix. The joint representations can be used in downstream tasks like VQA (left) or classification (right).

$$H_{G_s} = \Psi_\theta\big(X_s, A_s, E_s\big) \in \mathbb{R}^{|V_s| \times D} \quad \text{(1a)}$$

$$H_{G_t} = \Psi_\theta\big(X_t, A_t, E_t\big) \in \mathbb{R}^{|V_t| \times D} \quad \text{(1b)}$$

where $D$ is the embedding dimension. For choice of GNN, there are a few variants to pick from given the considerable recent research in geometric learning and GNNs [32, 49, 43, 20]. At a high-level, all these networks follow a *neural message passing scheme* [16], and iteratively update vectorial node representations by aggregating representations of its neighbours [7, 18]. This involves two key operations: 1. AGGREGATE, and 2. MERGE. A node's representation, $h_v^k$, after k iterations capture structural information within a $k$-hop neighbourhood in the graph. Formally:

$$a_v^{(k)} = \textbf{AGGREGATE}^{(k)}\bigg(\Big\{h_u^{(k-1)} : u \in \mathcal{N}(v)\Big\}\bigg), \quad h_v^{(k)} = \textbf{MERGE}^{(k)}\bigg(h_v^{(k-1)}, a_v^{(k)}\bigg),$$

$$\text{(2a-b)}$$

The final layer representation, $h_G^{(K)}$, is obtained using a READOUT function that combines node features from the final iteration to obtain entire graph's representation $h_{G_s}$ or $h_{G_t}$ as:

$$h_G = \textbf{READOUT}\big(\big\{h_v^{(K)} \big| v \in G\big\}\big) \quad \text{(3)}$$

For group equivariant representation of object attributes, relationships, it is important to have the READOUT function as a permutation invariant function like summation or a more sophisticated

graph-level pooling function [57]. To that accord, we use $\Psi_\theta$ as a two-layer *graph isomosphism network* (GIN) [53] – a GNN variant that has theoretically maximal representational power under the Weisfeiler-Lehman (WL) [51] graph isomorphism tests to distinguish between graph structures. This obtains vectorial local node representations $H_{G_s}, H_{G_t}$ for our source and target graphs in eq 1.

Using $H_{G_s}, H_{G_t}$, we get a soft correspondence matrix, $\Phi$ between the nodes $v_s \in |V_s|$ and $v_t \in |V_t|$ by: $\hat{\Phi} = H_{G_s} H_{G_t}^T \in \mathbb{R}^{|V_s| \times |V_t|}$ – where the $i$-th row vector $\Phi_{i,:} \in \mathbb{R}^{V_t}$ is a probability distribution over potential correspondences to nodes in $G_t$ for $\forall_i \in V_s$. Intuitively, one can think of this distribution as being a likelihood score to measure the goodness of matches between nodes in the two different graphs. To get a discrete correspondence distribution between the source and target nodes, we then apply 'sinkhorn normalization' [45] to the correspondence matrix to satisfy *rectangular doubly-stochastic* [1] matrix constraints such that $\sum_{j \in V_t} \Phi_{i,j} = 1, \ \forall i \in V_s$ and $\sum_{i \in V_s} \Phi_{i,j} \leq 1, \ \forall j \in V_t$.

$$\Phi = \mathbf{sinkhorn}\big(\hat{\Phi}^{(0)}\big) \in \big[0,1\big]^{|V_s| \times |V_t|} \tag{4}$$

Given $\Phi \in \mathcal{R}^{|V_s| \times |V_t|}$, we can project a function in the source (text) latent space $L(G_s) \in \mathcal{R}^{|V_s|}$ into the target (image) latent space $L(G_t) \in \mathcal{R}^{|V_t|}$ using

$$\vec{h}_t' = \Phi^T \vec{h}_s, \quad \vec{h}_s' = \Phi \vec{h}_t, \tag{5a-b}$$

The final joint multimodal representation $h_{s,t} \in \mathbb{R}^{2D}$ is obtained by concatenating $[h_s, \vec{h}_s']$, $\vec{h}_s'$ is target node $h_t$ projected in source latent space.

### 3.2 Using the multimodal representation in downstream tasks

For our caption classification task, the output of the matcher $h_{s,t}$ can be fed to a fully connected layer with sigmoid activation for discriminative classification. Formally, we optimize the binary cross-entropy loss function: $\mathcal{L} = -y\log(\hat{y}) - (1-y)\log(1-\hat{y})$ – where $\hat{y}$ is the model prediction for caption correctness, and $y$ is the ground-truth label.

For the VQA tasks, the joint representation, $h_{s,t}$ is fed to an attention-based sequence to sequence (seq2seq) [5] model with an encoder-decoder structure. We use a bidirectional LSTM [24] as our encoder. At time step $i$, the encoder takes a question $q_i$ containing variable length word tokens with added padding and the multimodal $G_s, G_t$ representation $h_{s,t}$ as input. The concatenated input $x_i \leftarrow [q_i; h_{s,t}]$ is encoded to obtain encoding $e_i$:

$$e_i = \big[e_i^F, e_i^B\big], \quad \text{where} \quad e_i^F, h_i^F = \text{LSTM}\big(\psi_E(x_i), h_{i-1}^F\big), \quad e_i^B, h_i^B = \text{LSTM}\big(\psi_E(x_i), h_{i+1}^B\big) \tag{6}$$

– where $\psi_E$ is the trained encoder embedding, and $(e_i^F, h_i^F)$, $(e_i^B, h_i^B)$ are the outputs and hidden vectors of the forward and backward networks at time step $i$.

The decoder is a similar LSTM that generates a vector $o_t$ from the previous token of the output sequence $y_{t-1}$. $o_t$ is then fed to an attention layer to obtain a context vector $c_t$ as a weighted sum of the encoded states by:

$$o_t = \text{LSTM}\big(\psi_D(y_{t-1})\big), \quad \alpha_{ti} \propto \exp\big(o_t^\top W_A e_i\big), \quad c_t = \sum_i \alpha_{ti} e_i \tag{7}$$

– where $\psi_D$ is the trained decoder embedding. Lastly, the decoder output and context vector is fed to a fully connected layer with softmax activation to obtain a distribution over the predicted program sequence token $y_t \leftarrow \text{softmax}(W_o[o_t, c_t])$. The predicted program sequence is used by a symbolic executor to answer a question for the VQA task.

## 4 Experiments

We empirically test our MGN model on two different tasks involving multi-modal inputs. For our first task, we consider a binary classification setup where the model has to predict whether a text caption is true of a given image. For the second, we evaluate systematic generalization using the recently proposed CLOSURE challenge [6]. The first task tests the model's ability to handle compositional variation in the image space, while the second tests its adaptation to variation in linguistic constructs.

## 4.1 Task 1: Caption Truth Prediction

In this task, given an image and a caption, the model has to predict if the caption is true (T) or false (F) in the context of that image. Figure 3 provides an example instance of this task.

We use images from the CLEVR dataset [28] and use their template generator to produce captions that are both true and false. The generation engine uses programmatic templates based on object attributes, logical predicates, and object reference predicates outlined in [28]. Templated question generation allows us to create complex questions, testing various aspects of scene understanding.

The original dataset contains 1M questions generated from 100k questions with 90 question template families that can be broadly categories into five question types: count, exist, numerical comparison, attribute comparison, and query. Note that the question types distribution is not uniform. Each generated question, **q**, is paired with an answer label, index of corresponding image, and a set of ground-truth modular program sequence $\{m_1, ..., m_n\} \in \mathcal{M}$, such that $\mathcal{M} \subseteq \mathcal{F}$, when applied gives the ground truth answer. Here $\mathcal{F}$ is a predefined environment specific function catalog. We delegate further details of this dataset to the appendix A.

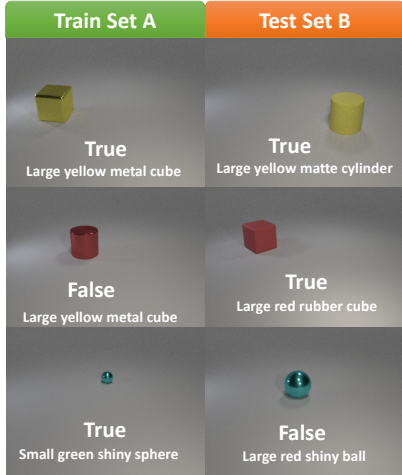

Figure 3: Sample instances for the Caption Truth Prediction task. Test set B contains object-attribute combinations that are unseen in Train set A.

**Variation in visual concepts** We train models on a subset of the available image object attribute values. During testing, the object attribute values are swapped and models are evaluated on whether they can detect the novel composition of attribute values without performance degradation. This tests whether the models are able to learn composite image features. The setup is aligned with the earlier proposed CLEVR-CoGenT dataset [28]. We synthetically generate two image datasets – A and B – with each image containing just one object. In dataset A, all **cubes** are gray, blue, brown, or yellow and all **cylinders** are red, green, purple, or cyan; in B, cubes and cylinders swap color palettes. Unlike CoGenT, which focuses of image attribute generalization, this task focuses on linguistic composition by applying more stringent truthiness test. Here we exhaustively test the variants of a true caption, by generating captions with all permutations and cardinality of object attributes.

The text input is an image caption describing the object by its attributes. The answer is a label 'True' or 'False' indicating the correctness of the caption.

### 4.1.1 Training

**Baselines.** For this task, we use baseline models – **CNN+LSTM** and **CNN+LSTM+SA** from Johnson et al.[28]. Both use a pretrained ResNet-101 CNN architecture for encoding the image, and a two-layer LSTM with 512 hidden units per layer for encoding the captions using learned 300-dimensional word embeddings. The (flattened) image features and encoded question are concatenated and passed to a two-layer MLP with 1024 units per layer, with ReLU nonlinearities after each layer. For the stacked-attention (SA) variant, instead of concatenation, the representations are fed to two consecutive Stacked Attention layers [54] with 512 units per layer. The output is a 512-dimensional vector that is fed to a linear layer to predict answer.

All models were trained using Adam with a learning rate of $5 \times 10^{-4}$, a batch size of 64 for a maximum of 360k iterations, with early stopping based on validation accuracy.

**MGN.** For our MGN model, we use a simple two-layer GCN [32] model trained in a supervised fashion. Using the parser[40] we convert the caption and image scene into $G_s, G_t$ and get corresponding graph data embeddings as: $(X_s, A_s) \leftarrow G_s, (X_t, A_t) \leftarrow G_t$ – where $X \in \mathbb{R}^d$ is a vector embedding of the graph nodes of dimension 'd' and $A$ is the adjacency matrix denoting edge indices. Then with the source, s and target, t as a pair-data sample input, and caption truthiness boolean as our

label, we use PyTorch Geometric [13] to train our **MCN-GCN** model. A learning rate of 0.01 with weight decay $5 \times 10^{-4}$ was used with the cross-entropy loss function.

### 4.1.2 Results

The findings of this simple focused test reveals (Table 1) our approach is indeed conducive to systematic generalization for image attributes.

| Model | Dataset A (seen) | Dataset B (unseen) |
|---|---|---|
| CNN+LSTM | 99.3 (0.5) | 77.5 (1.3) |
| CNN+LSTM+SA | 99.4 (0.3) | 79.7 (0.7) |
| MGN-GCN (ours) | **99.6 (0.3)** | **98.2 (1.1)** |

Table 1: Results on *Caption Truth Prediction*. Accuracy and standard deviation (in parenthesis) are computed over 3 random data splits. Dataset B contains novel unseen combinations of objects and attributes to test generalization ability.

The baseline models are learning image features with a joint embedding of the entire caption, for e.g., they are learning 'a red cylinder', and a 'brown cube' during training, and can't detect 'red cube' and 'brown cylinder' during testing.

## 4.2 Task 2: CLOSURE

In this second setup, we train models on the original CLEVR question templates, and test them against questions provided by CLOSURE templates [6]

**CLOSURE questions.** The question templates in CLOSURE systematically use primitive language constructs to create unseen combinations in the generated questions. Seven different templates are used, targeting one of the five broad CLEVR question types (`count`, `exist`, `numerical comparison`, `attribute comparison`, and `query`). For elucidation, we discuss one template (`and_mat_spa`) here. Please see the appendix B for further illustrations.

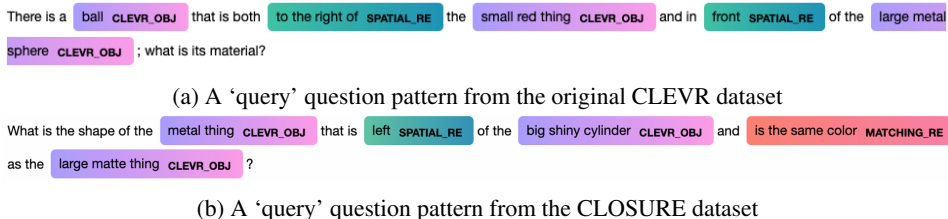

(a) A 'query' question pattern from the original CLEVR dataset

(b) A 'query' question pattern from the CLOSURE dataset

Figure 4: The `and_mat_spa` question template. N.B. the color codes and tags used are for illustrative purposes only, the model sees unannotated text as input.

In the original dataset, all object attribute *query* questions with logical conjunctions (`and`) uses spatial predicates: {`left`, `right`, `front`, `behind`}$\in$ `SPATIAL_RE` for relational reasoning between objects. In Fig.4, the `and_mat_spa` generated questions contain the same type (attribute *query* with logical conjunction 'and'), however, the relation reasoning employs both spatial and 'matching relations' (`MATCHING_RE`) where object references are made using matching predicates: "same {`shape`, `color`, `material`, `size`}". The rest of the CLOSURE templates employ the same trick to systematically rearrange known constructs to compose questions of the same type.

### 4.2.1 Training

We train the MGN components (section 3.1) in two phases: supervised pretraining and end-to-end fine-tuning using REINFORCE [52].

**Supervised pre-training.** For the pretraining data, we use 270 stratified question samples from each of the 90 question families in CLEVR. For obtaining parsed source and target graphs, $G_s$, $G_t$,

| Model | Mean Accuracy |
|---|---|
| FiLM | 56.91% |
| PG-Tensor-NMN | 64.51% |
| PG-Vector-NMN | 75.57% |
| NS-VQA | 77.19% |
| MAC | 73.80% |
| MGN-e2e (ours) | **80.87**% |
| MGN-e2e (w/o RL finetuning) | 72.18% |
| MGN-e2e (w/o pre-training) | 65.25% |

Table 2: Mean accuracies of baselines and our model (MGN-e2e) on CLOSURE. Bottom two rows represent ablations of our full model. Both supervised pre-training and RL fine-tuning play an important role in effective learning.

we use the CLEVR Parser library [40], whose components are detailed in 3.1.1. For the initial feature embeddings of $G_s, G_t$, we use the 'en_core_web_sm'[3] LM trained on the OntoNotes corpus [50]. The LM we use here is a design choice – alternatives can be one-hot vectors in the simplest case, or transformer/BERT based large LM [14, 42] embedding.

The graph matcher component with GNN $\Psi_\theta$ is pretrained in a supervised, discriminative fashion using ground truth correspondence labels $\pi_{gt}(.)$, by optimizing for correct correspondence score (between source $V_s$ and target $V_t$ nodes) using NLL objective: $\mathcal{L} = -\sum_{i \in V_s} \log \left( \Phi_{i, \pi_{gt}^{(i)}} \right)$. For training the GNN, we use the PyTorch Geometric library [13]. The seq2seq program generator is pretrained with questions and ground-truth program sequences for 25000 iterations. Both the encoder and decoder have hidden layers with a 256-dim hidden vector. We set the dimensions of both the encoder and decoder word vectors to be 300, and the multimodal graph vector representation to be 100.

**Reinforcement learning.** In the fine-tuning phase, we use REINFORCE [52] to train the entire flow end-to-end using the final answer correctness as the reward objective. We use a learning rate of $1 \times 10^{-5}$ and a batch size of 64 for a maximum of 1,000,000 iterations.

### 4.2.2 Results

We compare our approach to five other baselines discussed in [6]. Table 2 shows the mean accuracies over the 7-templates compared with our best performing MGN model. Note that the results reflect test on 'zero-shot generalization' only.

Mean accuracies do not capture the compositional abilities of some the stronger models. For e.g, the NS-VQA[55] and the MAC model introduced by Hudson et al. [27] perform strongly overall, however, looking at the performance breakdown by template in Figure 5, we see these models perform poorly on logical relationships (`and_mat_spa`, `compare_mat`). MGN, on the other hand, performs well across all 7 templates.

Figure 5 shows our model's mean accuracy over all the 7 CLOSURE templates against the baselines.

**Ablation tests.** We also perform ablation analyses of our MGN model with respect to the pre-training and fine-tuning phases used (Table 2). We observe that both the supervised pre-training and the RL fine-tuning are crucial to obtaining good performance. We report more ablation studies and discuss them in detail in the Appendix.

## 5 Conclusion

In this paper, we proposed Multimodal Graph Networks (MGNs) to solve the problem of compositional generalization for visual question answering. MGNs induce a tighter coupling between concepts in different modalities (e.g. images and text) by learning joint graph-based representations,

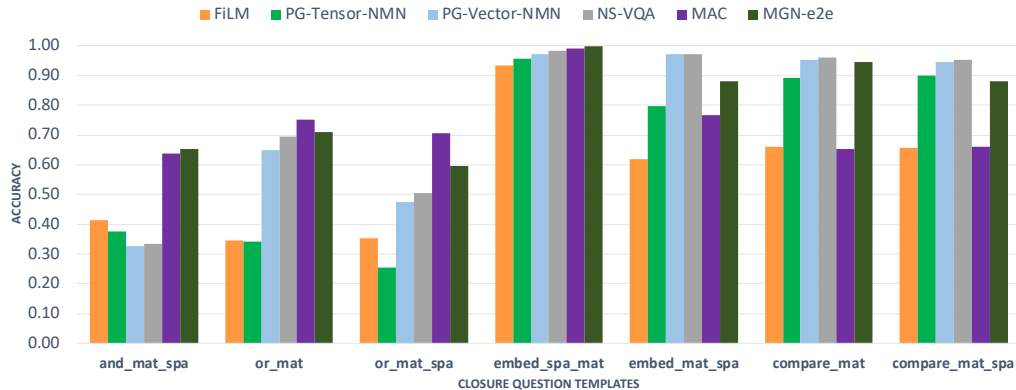

Figure 5: MGN vs. Baselines on individual CLOSURE templates. MGN performs consistently well on almost all templates while baselines like MAC and NS-VQA do well on a subset while performing worse on others.

which are compositional in nature and allow us to capture entities, attributes and relations in a scalable manner. Our method predicts a factor correspondence matrix to implicitly induce factor nodes between concepts that are related in the two modalities. Empirically, our approach results in state-of-the-art results on the recently introduced CLOSURE dataset, improving on the mean overall accuracy by 4.77% over previous approaches. Our model also performs consistently well across different types of compositions of linguistic constructs while prior methods only succeed on a subset of them. The current results for MGN are promising on *synthetic* image data with limited (constrained) language vocabulary. Future work can tackle the challenge of extending MGN's compositional characteristics to more natural image datasets by scaling up the graph parser to large number of object categories and handling larger vocabularies in the text.

## Broader Impact

Multimodal reasoning methods that can generalize to novel compositions of linguistic constructs are key to developing more intelligent systems that can reason and ground language in various contexts. Though this work is evaluated on synthetic datasets with generated images, we believe that graph-based techniques can provide smoother scaling and better handle challenging fine-grained multimodal reasoning than alternative approaches. We envision this work to enable improvements on various downstream applications like instruction following, autonomous navigation, and robotic control. However, improved multi-modal reasoning can also result in systems that can be harmful to society (e.g. surveillance systems) if misused. There is also the aspect of various forms of bias that the model may pick up if trained with data that only represents a sub-set of the phenomena in the world. In the current form, MGN does not algorithmically account or correct for bias in multimodal data even though it may provide a useful starting point due to its use of explicit graphs – this can be another direction for future research.

## Acknowledgements

Raeid Saqur was supported by the Fulbright Scholarship program, and the RBC Fellowship. Resources used in preparing this research were provided, in part, by the Province of Ontario, the Government of Canada through CIFAR, and companies sponsoring the Vector Institute www.vectorinstitute.ai/#partners.

## Footnotes

[3]`https://spacy.io/models/en#en_core_web_sm`

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
