[Supplementary Material · appendix.pdf]

# Appendices

## A    CLEVR Domain-Specific Language (DSL) and Functions Library

We tabulate the function library for the CLEVR VQA [28], and the corresponding signatures. The functions and signatures were kept unaltered from the original specifications [4] for direct comparison. Figure 6 shows a topological overview the dataset with sample image, corresponding questions, program chains using the function catalogue.

Figure 6: Overview of the CLEVER dataset

### A.1    CLEVR Object Attributes

Table 3: Variability in object attributes

| Attributes | Symbols | Values | Cardinality |
|---|---|---|---|
| Color | <C> | "gray", "red", "blue", "green", "brown", "purple", "cyan", "yellow" | 8 |
| Size | <Z> | "small", "large" | 2 |
| Material | <M> | "rubber", "metal" | 2 |
| Shape | <S> | "cube", "sphere", "cylinder" | 3 |

### A.2    CLEVR Domain Specific Language (DSL) Function Catalog

Table 4 in the following page lists the modular programs defined in the CLEVR domain specific language (DSL)

## B    CLOSURE Dataset: Systematic Generalization Tests for CLEVR Models

The original CLEVR dataset questions can be categorized into five broad question types (`count`, `exist`, `numerical comparison`, `attribute comparison`, and `query`). The question templates used to generate these questions use referring expressions (RE) to refer to an object or a set of objects. An atomic referring expression (`simple_re`) refers to an object using its attributes, for e.g. *'the big red cube'*. A complex referring expression (`complex_re`), uses multiple conditionals to refer to an object or objectset using the spatial relationships between objects (`spatial_re`) or their attribute similarity (`matching_re`). For e.g., '*<obj> that is {left, right, in front, behind}[of] <simple_re>*' is a `spatial_re` referant; and a referent using matching predicate like '*is the same size as <simple_re>*' is an e.g. of a '`matching_re`'. Finally, `logical_re` are questions where two referents are combined using conjunctions: `and` or `or`.

The CLOSURE questions [6] uses seven templates to create CLEVR questions, but with the patterns of these primitive referents altered. To illustrate, consider the 'comparison' questions in CLEVR (e.g. 'Is the size of the red cube left of the blue cylinder same as the metal ball?'), which only

| Operation | Signature | Semantics |
|---|---|---|
| Scene | () ⟶ ObjectSet | Return all objects in the scene. |
| Filter | (ObjectSet, Obj) ⟶ ObjectSet | Filter out a set of objects having the object-level concept (e.g., red) from the input object set. |
| Relate | (Object, Rel) ⟶ ObjectSet | Filter out a set of objects that have the relational concept (e.g., left) with the input object. |
| Intersection | (ObjectSet, ObjectSet) ⟶ ObjectSet | Return the intersection of two object sets. |
| Union | (ObjectSet, ObjectSet) ⟶ ObjectSet | Return the union of two object sets. |
| Query | (Object, Attribute) ⟶ Obj | Query the attribute (e.g., color) of the input object. |
| Exist | (ObjectSet) ⟶ Bool | Query if the set is empty. |
| Count | (ObjectSet) ⟶ Integer | Query the number of objects in the input set. |
| Equal_<attr> | (attr1, attr2) ⟶ Bool | (Attribute Equal) Query if the argument attributes are equal. |
| Equal_Integer | (integer1, integer2) ⟶ Bool | (Counting Equal) Query if the number of objects in the first input set is the same as the one of the second set. |

Table 4: Pertaining to section A.2: all operations in the domain-specific language for CLEVR VQA.

uses `spatial_re` as referent to the final matching predicate. The corresponding closure templates (compare_mat, compare_mat_spa) recombines these questions using known constructs (i.e. the matching REs and comparision questions), in a novel composition. For e.g. the `spatial_re` as referents are replaced with `matching_re` referents in the question.

Is the size of the [sphere CLEVR_OBJ] in [front SPATIAL_RE] of the [yellow matte ball CLEVR_OBJ] the same as the [big red cylinder CLEVR_OBJ] ?

(a) An 'existence' type question using original CLEVR question template

There is another [tiny matte object CLEVR_OBJ] that [is the same shape MATCHING_RE] as the [big green matte object CLEVR_OBJ] ; does it have the [same color MATCHING_RE] as the [tiny cube CLEVR_OBJ] ?

There is another [ball CLEVR_OBJ] that [is the same material MATCHING_RE] as the [red sphere CLEVR_OBJ] ; does it have the [same size MATCHING_RE] as the [blue rubber block CLEVR_OBJ] ?

(b) An 'existence' type question using CLOSURE template `compare_mat`

Figure 7: `compare_mat_spa` question template example showing the compositional make of *'existence'* type questions in CLOSURE juxtaposed with the original datset. Note the referent using spatial relationship (highlighted as yellow) between objects in 7a has been replaced with matching predicate referents (highlighted as salmon) in 7b

.

Figure 8 illustrates this compositional shift of known constructs in a `logical_re` question where conjunction 'or' is used to combine multiple, nested referring expressions for 'count' type of questions.

In all of the preceding examples, the CLOSURE questions do not introduce any unknown construct that the model hasn't seen apriori during training. The templates only changes the pattern of occurrence in a novel composition, thus testing the model's ability to systematically generalize to

How many [things CLEVR_OBJS] are either [small green objects CLEVR_OBJS] in [front SPATIAL_RE] of the [small purple cylinder CLEVR_OBJ] or [large metallic things CLEVR_OBJS] that are behind the [SPATIAL_RE] [red matte thing CLEVR_OBJ] ?

(a) A 'count' type question using original CLEVR question template

How many [things CLEVR_OBJS] are [cylinders CLEVR_OBJS] that are behind the [SPATIAL_RE] [large purple metal thing CLEVR_OBJ] or [purple cylinders CLEVR_OBJS] that [are the same size MATCHING_RE] as the [cyan thing CLEVR_OBJ] ?

(b) A 'count' type question using CLOSURE template `or_mat_spa`

Figure 8: `or_mat_spa` question template example showing the compositional make of *'count'* type questions in CLOSURE juxtaposed with the original datset. Note the last referent using spatial relationship (highlighted as yellow) between objects in 8a has been replaced with matching predicate referents (highlighted as salmon) in 8b

.

novel compositions of known constructs [14]. Please see the original paper for detailed illustrations [6].

## C Compositional Generalization in Language

For the purposes of this paper – and from a general machine learning, NLP context – we use '*compositional*' and '*systematic*' generalization interchangeably, and formalize the problem.

Compositional generalization in language means the ability to form arbitrary combinations of atomic or primitive components of language from a fixed set of primitives or known components [10, 34]. It is a hallmark of human cognition [Minsky, 1986, Lake et al. 2017] and imperative component of language acquisition [Biemiller, 2001]. A system's ability to model language compositionality has been a staple for criticisms of AI and in debates ranging from 1980s connectionist-classicist approaches to modern neural vs. symbolic approaches [30, 12, 44, 14].

There has been renewed vigour in this topic in recent NLP and ML work [31, 35, 34][Lake & Baroni, 2018], due to its potential impact in various ML sub-domains (see Section 5). Since this paper builds on a multimodal symbol grounding approach [21], we streamline the problem setup to a multimodal context, specifically visual reasoning or visual question answering (VQA). Current SotA models in this domain, with the capability to solve intricate and complex questions, suffer catastrophic failures even with minor shifts in underlying compositional language patterns in constrained domain-specific language (DSL) setups.

Consider the following questions and their corresponding canonical logical forms [56, 8] using $\lambda$-calculus [11]:

| | |
|---|---|
| 1 | What is the shape of the thing that has the same color as the rubber cube? |
| | $\lambda y.\texttt{query\_shape}(y, \lambda x.\texttt{equal\_color}(x, y))$ |
| 2 | There is a ball left to the green cylinder; what is its material? |
| | $\lambda y.\texttt{query\_material}(y, \lambda x.\texttt{relate\_left}(x))$ |

Table 5: Training data observed by a model.

Compositional desiderata dictate that having learned how to answer the above questions using functional primitives, the model should be able to answer compound questions that can be answered using a composition of the learned functions, i.e., to achieve zero-shot generalization to distributionally shifted primitive function patterns.

| 3 | What is the shape of the metal thing that is left of the big shiny cylinder and is the same color as the large matte thing? |
|---|---|
| | `λy.query_shape(y, filter_metal(λx.relate_left(x) ∧` `λx.equal_color(x)))` |

Table 6: Test data observed by a model w/o apriori exposure: 'zero-shot generalization' task.

# D  Additional Experiments

## D.1  MGN variants

We also evaluate ablation tests on our MGN model. Table 7 shows the mean accuracies of MGN variants we tested. The main variant to our presented variant: '**MGN-e2e**' is '**MGN-fixed**' – where the '*graph matcher*' component is taken out of the end-to-end training pipeline and pretrained separately, i.e., during fine-tuning phase, we do not update the matcher GNN's parameters. Apart from that, the flow remains the same. We also notice that the choice of training samples trained on (both pretrain and finetune), has significant impact on the results.

For pretraining, we do: a) k-stratified sampling from each of the 90 question templates – where $k \in \{50, 270\}$. Concretely, for k=50, we used $50 \times 90 = 4500$ training samples for pretraining, and likewise for k=270; b) Random sampling of a subset of training samples.

For naming convention in the Table 7, we use the format: `"MGN-[fixed|e2e]-<pretrain-samples>-<number of fine-tuning steps>"`. 'S' in pretrain denotes stratified sampling by template.

| Model | Mean Accuracy |
|---|---|
| MGN-fixed-50pretrain-20kiters.finetune | 64.79% |
| MGN-fixed-375k.pretrain-full.finetune | 68.11% |
| MGN-fixed-700k.pretrain-full.finetune | 72.18% |
| MGN-e2e-50Spretrain-475k.samples.300kiters.finetune | 58.59% |
| MGN-e2e-270Spretrain.475k.samples.200kiters.finetune | 49.54% |
| MGN-e2e-50Spretrain.475k.full.finetune | 60.81% |
| MGN-e2e-270Spg.475k.full.finetune | 62.97% |
| MGN-e2e-270Spretrain-300kpg | 65.25% |
| MGN-e2e-270Spretrain-600kpg | 58.94% |

Table 7: MGN variants' mean accuracies across the seven CLOSURE templates.

## D.2  Caption Truth Prediction: Multi-object Tests

We extend the '1obj' image attribute compositionality tests with multiple objects against baseline models in 4.

# E  Graph Parsing from Questions and Image Schene Graphs

This section further illustrates the graph parsing flow by the graph parser detailed in section 3.1.1 by providing additional examples.

Figure 9: MGN Variants on CLOSURE templates

(a) An Image from the CLEVR dataset

How many spheres in front of the red metal cube and to the right of purple sphere

(b) Graph representation of an example question on the image in 10a: $\mathbf{G_s}$

(c) Corresponding graph representation of the parsed image scene graph: $\mathbf{G_t}$

(d) Visualizing both the image and the question in a bipartite graph. Orange nodes represent the text nodes, and the aquamarine nodes represent the image nodes: $\mathbf{G_u}$

## Footnotes

[4]https://github.com/facebookresearch/clevr-dataset-gen/blob/master/question_generation/metadata.json