[Reviews · NeurIPS 2020]

Review 1

Summary and Contributions: The paper proposed a new VQA model that aims to achieve better compositional generalization, e.g. to new combinations of objects and properties that didn’t appear in the training set. The authors first build a graph representation that integrates language and vision and then use graph network techniques to compute a factor correspondence matrix which is finally turned into a symbolic program that predicts answers. Update: I thank the authors for their response. After reading it I unfortunately still stay in my position about the paper: the main weaknesses in my view are first the novelty of the work which seems to me to be limited and the fact that it was only evaluated on artificial data with relatively small number of possible generalization combinations. While it is true that the question processing in this work is different than e.g. NSM, comparing to both this and NS-VQA, the core idea of turning the data into scene graphs is not novel and therefore I would like to keep my original score.

Strengths: The model description is clear and the visual depiction in the first figure is very useful and adds to its clarity. For the CLEVR dataset (and CLOSURE questions on it) multiple experiments are being performed and stronger results than prior works are achieved. The paper is well written and easy to follow.

Weaknesses: * In terms of novelty, note that both the motivation for the model as well as the initial parts of it hold similarities to some prior works. See detailed description in the relation to prior work section. * I would be happy to see results about generalization not only for the CLEVR dataset, but also for natural images datasets where there is larger variance both in the language and visual complexity. There are multiple datasets for generalization in VQA that can be used for that such as CP-VQA and also some splits of GQA. For the CLEVR dataset, the model is basically based on using an object detector to recognize the objects and their properties and build a semantic graph that represents the image. While other approaches that are compared to for this task use object detectors as well, there are many approaches for CLEVR (such as the Neural Module Network, Relation Network, MAC and FiLM) that do not use such strong supervision and therefore the comparison between these approaches in the experimental section is not completely valid. For better comparability, I would be interested to see generalization results when these models are also being fed with at least object-based bounding-boxes representations instead of the earlier commonly used spatial features, as is very common in VQA in the last years (see bottom-up attention networks). * This can be open for debate but I personally believe that the need for reinforcement learning for a static VQA task may be a potential weakness making the approach less data efficient and harder to train the models that use gradient descent.

Correctness: The claims in the paper are corroborated by experiments and the model description is clear and correct.

Clarity: The paper clarity is good and it is well written and easy to follow.

Relation to Prior Work: Not completely. Both the motivation for the model as well as the initial parts of it hold similarities to some prior works,, e.g. learning by abstraction: the neural state machine that was presented at last NerurIPS, which builds a multi-modal graph as well and uses graph-networks message-passing techniques to perform iterative reasoning, showing better generalization along multiple axes. The usage of semantic concepts to relate between language and vision is also quite similar. There are also some similarities to the NS-VQA model. It would be good to discuss similarities and differences to these models and other relevant prior works in the related work section. However, it is important to mention that the latter parts of the model such as the factor correspondence matrix and the symbolic program are more different from prior works that I’m aware of in that direction.

Reproducibility: Yes

Additional Feedback: * Adding more details about graph isomorphism networks and sinkhorn normalization in the model section in page 4 will be useful. * More details about the symbolic programs that are generated and examples of how the programs look in the model section in page 5 will also be beneficial. * There is a chance I’m wrong on that but if I understand correctly, the first experiments are being performed on a set of true/false statements automatically constructed by the authors for the CLEVR dataset, in order to measure compositional reasoning. I’m wondering why not to use the standard CLEVR questions to measure that? I believe that as long as the newly introduced data doesn’t provide or allow testing new aspects or tasks, it’s better to use common data for better comparability to prior approaches. In addition, the standard CLEVR questions allow further variability in answers and reasoning skills needed than true/false statements and is carefully constructed to mitigate shortcuts and biases and so may be a better benchmark to use for the task of compositional reasoning. * However, the table on page 6 sounds like the standard CLEVR CoGenT task is used. If so, when are the new True/False generated statements that are discussed in the bottom part of page 5 are used? * For the CLOSURE dataset, the questions are described with color coding and tags for different words, such as objects and relations. Just to be certain about that, are these explicitly fed into the model or used for supervision in any way or does the model see unannotated language? Small comments * Line 46 there is an unnecessary comma at the end of the sentence. * For the sentence in 248: they are rather than it’s


Review 2

Summary and Contributions: Presents a new GNN approach for VQA and caption verification that does well on the CLOSURE data

Strengths: Using graph representations to provide more compositional/systematic generalization is an interesting and important problem. The paper presents a reasonably novel and interesting method using graph representations of both language and images to sole multimodal problems like image/caption verification and VQA. Shows improved results on the recently introduced CLOSURE dataset derived from CLEVR.

Weaknesses: Presented results are all on synthetic image/language datasets. Why not present results on the GQA dataset which provides compositional questions on real images? Or HumanCLEVR which provides human paraphrases for the artificial NL questions in CLEVR. It was unclear to what extent the authors' method uses additional supervised training data that are not exploited by the methods compared to. Particularly, what (supervised) language data/knowledge is used by the language and image parser? Do what extent do the competing methods exploit these resources? It seems that the VQA system produces symbolic programs that are then executed and that it uses supervised programs during training the decoder. Do the competing methods compared to exploit this supervision also. It would be clear to what extent the comparison are "fair" in that they all exploit the same training resources. I'm not familiar with CLOSURE, are the results presented SOTA? Have other GNN and neurosymbolic methods referenced been applied to this dataset, comparisons to them seems necessary since they would more naturally handle compositionality. The broader impacts section does not discuss potential NEGATIVE impacts as requested in the instructions for this section. Minor: the citation for the "neural message passing algorithm" (line 106) is blank

Correctness: Paper seems technically sound, except see issues above on the experimental comparisons.

Clarity: Paper is fairly well written and clear.

Relation to Prior Work: Related work sees comprehensive and discussed to my knowledge

Reproducibility: Yes

Additional Feedback:


Review 3

Summary and Contributions: The paper proposes to handle compositional generalization in visual question answering using multimodal graph networks. Specifically, it extracts the graphs from the image (object and its attributes) and the text, embeds them using graph neural networks, and computes the crossmodal interaction to obtain a multimodal representation (MGM-e2e) for the downstream tasks. Experimental results on caption truth prediction and VQA on CLOSURE dataset show superior performances over existing methods.

Strengths: (S1) The proposed approach to use graphs to capture the intramodal dependencies followed by a crossmodal interaction is intuitive and novel to achieve compositional generalization.

Weaknesses: (W1) The technical challenge of Task 1 is extremely unclear. Given an image that contains exactly one object and a capture containing all four attributes, the goal is to classify if the caption rightly describes the image or not. Since the proposed approach depends on the parsed to get G_t (graph for text modality) and pretrained model to get G_s (graph to extract visual attributes), the goal now becomes to just check if the four attributes (from text and image) match or not. This becomes a trivial problem of semantic matching of labels and does not necessitate the complex graph fusion that is proposed. Thus, it raises the question whether the results in Table 1 can be considered as sufficient evidence as the complexity appears to be only superficial. (W2) The experimental comparison appears to be unfair. In Table 2, the performance of MGN-e2e is ~81% (w/ RL finetuning) and ~72% without. Some of the competing methods perform better than MGN-e2e (w/ RL finetuning), for example, NS-VQA at 77%. Thus, begging the question whether the improvement is purely from gradients using reinforcement learning as opposed to the proposed architecture. Since one piece of evidence suggests so (77% of NS-VQA vs 72% of MGN-e2e w/ RL), I suggest the authors to also fine-tune NS-VQA using REINFORCE for a fairer comparison between the methods.

Correctness: Due to above issues (W1 and W2), the experimental evidence to show the efficiency of the proposed system is inadequate. Without further experiments, the source of improvement in the performance is not clear thus questioning the effectiveness of the current approach.

Clarity: Overall, the paper is easy to read and follow without difficulties except for the follow minor issues: (a) L106 -- Missing reference (b) Is A_t fixed to be the adjacency matrix of a disjoint graph? If yes, please state it clearly. (c) L161: What are H_s and H_t? Are they supposed to be H_G_t and H_G_s? (d) Figure 3: Train Set A, 3rd example, Green -> Cyan or True -> False?

Relation to Prior Work: The manuscript does a reasonably good job to place the current work in the context of prior works.

Reproducibility: Yes

Additional Feedback: ------ I thank the authors for their response 1. The response correctly pointed out the technical detail about prior work (NS-VQA also uses REINFORCE) that I missed earlier in my evaluation. The clarification definitely addresses my concern regarding experimental validation for CLOSURE that formed the basis for my earlier rating. 2. The authors mention that task 1 is for efficacy and not complexity. Thus, it stands as a nice sanity experiment about the model and not particularly throwing light on comparison with respect to prior work. 3. The issue of lack of results in real world datasets is still present (as observed by other reviewers as well). Taking all the above points into consideration, I am happy to raise my rating to a "Marginal Accept" to truly reflect the merit of the paper (novel and interesting approach).


Review 4

Summary and Contributions: This paper introduces a method for compositional reasoning with synthetically generated image-and-text pairs. In the first phase, the method, called MGN, first parses the synthetic text string into a graph structure, then parses the synthetic image into a graph structure. In the second phase, it applies a graph matching algorithm that combines the two graphs into a single vector representation. The vector representation is used as input to downstream tasks, such as deciding whether the text string reflects the image content accurately. The system is compared against a few existing systems on the CLOSURE dataset and a modified version of the CLEVR dataset (for caption truth prediction). After the rebuttal and discussion I am leaving my score the same. My main concerns are still the missing comparison to NSM and evaluation on natural data.

Strengths: The system itself is clearly described and performs well on the more challenging evaluation settings than its comparisons.

Weaknesses: The main limitation of this work is evaluation on only synthetic data and missing relevant related work. The goal of this paper is to assess generalization to new compositional structures, which prior work has failed to do even on synthetically-generated data. However, there exist many image-text benchmarks that also target compositional generalization, including GQA (Hudson and Manning 2019), which includes real images but synthetic language, and several versions of VQA (such as VQA-CP (Agrawal et al. 2018) and VQA 2.0 (Goyal et al. 2017)), which include both real images and text. Both of these benchmarks specifically evaluate generalization to new compositional structures. Similar datasets which target complex, compositional natural language reasoning include NLVR (Suhr et al. 2017) and NLVR2 (Suhr et al. 2019). Evaluating on datasets with natural data components, whether it be images, language, or both, would make the paper much stronger. Even if the system is only evaluated on CLOSURE and CLEVR, these existing benchmarks should be referenced. Beyond these benchmarks, some relevant related work which is missed is listed below.

Correctness: The system is evaluated on two tasks -- a caption truth prediction task and a question-answering task. I had a few questions about the choice of baselines/comparison systems. First, I'm wondering why the systems compared against for the two tasks aren't the same. Truth prediction could be treated as a form of question answering, so the comparison systems described in Table 2 could also be evaluated for truth prediction. Second, there are relevant systems which are also designed to perform compositional reasoning about an image-text pair, including LXMERT (Tan et al. 2019), VilBERT (Lu et al. 2019), and NSM (Hudson and Manning 2019). In particular, NSM is quite similar to the proposed system, using graph neural networks derived from the image (unlike MGN, the text string is not parsed into a graph). How do these systems compare to MGN on both tasks? A discussion comparing NSM with MGN would also be beneficial. Last, the choice to evaluate caption truth prediction isn't clear. Why was this task chosen instead of using the original CLEVR task?

Clarity: The description of the model in Section 3 is relatively easy to understand. A few minor suggestions: * There seem to be some formatting errors throughout the paper (e.g., seems to be missing citation in line 106) * For Figures 1 and 2, the abstract graphs make the figure noisy and don't really contribute much as they aren't labeled (except the leftmost graph in Figure 1). A figure which shows exactly what the graphs would look like for a particular example (the figures would have to be larger) would be much more informative, rather than having abstract figures without labels.

Relation to Prior Work: There is discussion missing of relevant benchmarks and methods, including the benchmarks listed above (GQA, VQA-CP, VQA 2.0, NLVR(2)), and the methods listed above (LXMERT, VilBERT, NSM).

Reproducibility: Yes

Additional Feedback: The broader impact section should also discuss potential ethical considerations of the proposed system.

[Author Response · NeurIPS 2020]

We thank all the reviewers for their insightful questions, comments and commendations (novelty, clarity, performance).
**Common Clarifications:** **(CC1)** *Evaluation with other datasets (VQA-CP, GQA)* @R1, R2, R4: The main focus and
scope of the paper is to demonstrate the efficacy of MGN in generalizing to unseen combinations of known linguistic
constructs. Using datasets like CLEVR and CLOSURE allows us fine-grained control over the evaluation and analysis
of our primary objective. Additionally, we tried evaluating on GQA but the size of the dataset and unconstrained
language vocabulary proved to be computationally expensive. We believe our promising results on CLEVR/CLOSURE
should generalize to GQA as well, and believe that future work can build upon our results.

**(CC2)** *Task 1 motivation, complexity* @R3, R4: Motivation behind Task 1 is efficacy, not complexity [L46-49]. We use
the caption truthiness task to show the representational efficacy of MGN embedding for classification tasks, in addition
to VQA. While we presented the 1-object case for clarity and space, we also obtain 99% accuracy on multi-object (2, 3
objects) caption correctness task, and the original CLEVR-CoGenT (8 objects) VQA task.

**(CC3)** *Prior Work* @R1, R4: *VQA-CP, VQA2.0*: These datasets and corresponding models test *generalization* across
distributional shifts in train-test class labels. GVQA (from VQA-CP) builds on stacked attention networks (SAN).
SAN, LXMERT, and ViLBERT use a two-stream architecture to embed each modality separately, then fuse them
using attention-based interactions. Thus, they are orthogonal to MGN in problem setting and architecture. CLEVR
already incorporates non-uniform train-test labels, and SAN performs reasonably well (73.2%) on CLEVR (baseline
in NS-VQA). However, SAN and, by extension, GVQA architectures do not evaluate for, and generalize poorly on,
unseen object attributes (CLEVR-CoGenT) and linguistic structural pattern (CLOSURE) combinations. While the
NSM model also builds a scene graph from the image similar to our approach, they convert the question text into a
sequence of instructions for graph traversal. In contrast, we convert text into a graph as well and perform multi-modal
fine-grained matching. We show results using MAC (from the GQA authors, L291) with both CLEVR and GQA results.
MGN matches MAC on CLEVR tasks, and performs better on CLOSURE [Table 2].

@R1 **Weaknesses (W1)** *Novelty*: The novelty of our contribution arises from shortcomings of cited work, and
MGN addresses compositional generalization using multimodal graph fusion. Even if components of our model share
similarities to prior work, the entire system is itself novel. **(W2)** *Other datasets*: Please see [*CC1, CC3*]. *Object*
*detector supervision*: Applies only to MAC and FiLM baselines. Modifying MAC knowledge base input to feed
detector supervision would be a significant model change, due to its motivation and design. The 'PG' baselines (Table
2) incorporate FiLM in their model design, and NS-VQA builds on NMN's functional program modules approach.
Thus, they are fair proxies for FiLM, NMN with object detector supervision. **Prior Work**: We specifically discuss
neural approaches like NS-VQA [L63], in 'Related Work' (Section 2 [L56-67]). Also see *CC3*. **Feedback**: We will add
in further elaboration on GIN and Sinkhorn normalization. Fig. 6 and Table 4 in appendix A, B provide more details
about the symbolic program generation. The dataset used for Task 1 is aligned with standard CLEVR-CoGenT dataset
[L212] aimed at VQA – we modify templates to cater to questions with binary answer labels for classification. For
CLOSURE, our model sees unannotated language as input – the color codings and tags in Fig. 4 are illustrative. We
have corrected formatting errors.

@R2 **(W1)** *Results on real images*: Please see [*CC1*]. **(W2)** *Comparison fairness using parsers*: The image parser is
pretrained (on 4K CLEVR images) [L129, Fig. 2] using ground-truth object attributes and 3D coordinates (identical
to NS-VQA). The language parser is not trained, constructs text $(s) \rightarrow$ object graphs $(G_s)$ using rules-based entity
recognizer [L126]. **(W3)** *CLOSURE performance*: To the best of our knowledge, the results are SOTA (Table 2, and
Fig 5), and no other approaches have applied GNN and neuro-symbolic methods to CLOSURE. **(W4)** *Broader Impact*:
We will update with discussions on negative impacts and ethical considerations.

@R3 **(W1)** *Task 1 complexity*: Please see [*CC2*]. **(W2)** *NS-VQA comparison unfair due to RL finetuning*: We
respectfully disagree. NS-VQA uses identical RL fine-tuning, and all NS-VQA results/comparisons are with RL
fine-tuning. **Clarity** Minor clarifications - 5a: corrected in the camera-ready version. 5b: Yes. $A_t$ is fixed adjacency
matrix of a disjoint graph $G_t$ [L123-125]. 5c: Yes. In [L161], $H_s$, $H_t$ is supposed to be $H_{G_s}$, $H_{G_t}$ – we have corrected
this typo in revision. 5d: In Fig. 3, third e.g., Green -> Cyan.

@R4 **(W1)** *Evaluation on synthetic data*: Addressed in [*CC1*]. **(W2)** *Related work*: Please see [*CC3*]. **Correctness**:
**(C1, C3)**: Please see [*CC2*]. Task 1 is significantly harder than CLEVR since the model has to correctly answer all the
statements (10 per image) to get each instance correct. We believe the models in Table 2 might do well here also, but
our goal is to simply use task 1 as a demonstration of our model's ability ( 98% success). **(C2)** *Related systems*: Please
see [*CC3*]. **Clarity**: In Fig 1, the $G_t$ graph for 'small blue sphere' is intentionally unlabeled alluding to its irrelevance
to the question, as specifically elucidated in L92. Appendix E in supplementary materials shows exactly what graphs
would look like for a particular example. **Prior Work**: [*W2@R4*]. **Broader Impact** 8,11: [*W4@R2*].

[Meta-Review · NeurIPS 2020]

After the author response and discussion all reviewers recommend (weak) accept of this paper for its contributions including: - Significant improvements on the synthetic CLEVR/CLOSURE task - Overall novel and interesting method I accept the paper with the expectation that the author will improve and clarify the paper according the author response and suggestions by the reviewers, including discussion of related work. The main concern of the reviewers and I is that the paper limits their experimental evaluation to the synthetic CLEVR dataset. The authors are strongly encouraged to include results on a non-synthetic dataset (e.g. VQA-CP, NVLR/2, GQA - or subsets if necessary) in the final version, even if results in a negative result which could be analyzed by the authors.